# ANN-Based Fatigue Strength of Concrete under Compression

**DOI:** 10.3390/ma12223787

**Published:** 2019-11-18

**Authors:** Miguel Abambres, Eva O.L. Lantsoght

**Affiliations:** 1Num3ros, 1600-275 Lisbon, Portugal; 2Escola de Tecnologias e Engenharia, Instituto Superior de Educação e Ciências (ISEC), 1750-142 Lisbon, Portugal; 3Politécnico, Universidad San Francisco de Quito, EC 170157 Quito, Ecuador; 4Engineering Structures, Civil Engineering and Geosciences, Delft University of Technology, 2628 CN Delft, The Netherlands

**Keywords:** artificial neural networks, codes, compression, concrete, cyclic behavior, databases, fatigue

## Abstract

When concrete is subjected to cycles of compression, its strength is lower than the statically determined concrete compressive strength. This reduction is typically expressed as a function of the number of cycles. In this work, we study the reduced capacity as a function of a given number of cycles by means of artificial neural networks. We used an input database with 203 datapoints gathered from the literature. To find the optimal neural network, 14 features of neural networks were studied and varied, resulting in the optimal neural net. This proposed model resulted in a maximum relative error of 5.1% and a mean relative error of 1.2% for the 203 datapoints. The proposed model resulted in a better prediction (mean tested to predicted value = 1.00 with a coefficient of variation 1.7%) as compared to the existing code expressions. The model we developed can thus be used for the design and the assessment of concrete structures and provides a more accurate assessment and design than the existing methods.

## 1. Introduction

When concrete is subjected to cycles of compression, its strength is lower than the statically determined concrete compressive strength [1,2]. The practical implication of this mechanical property is that we need to consider a lower concrete compressive strength for structures subjected to cycles of loading, also called fatigue loading, such as bridges subjected to repeated traffic loads [3,4,5]. At the basis of the fatigue problem lie slow crack propagation [6] and creep [7]. In experiments, the behavior of a specimen can be characterized by the increase in strain over time, where a fast increase in strains is a precursor for fatigue failure [1].

The most fundamental approach to study fatigue is by isolating the different material contributions in the cross-section [8,9]. As such, the effect of fatigue on concrete under compression in a cross-section is studied separately by testing concrete cylinders under cyclic loading [10,11]. This fundamental knowledge together with information about the fatigue life of concrete under tension [12,13,14,15] and the fatigue behavior of reinforcement and prestressing steel [9,16,17] lies at the basis of studying the influence of fatigue loading in structural elements [18]. 

The knowledge about the fatigue behavior of materials under controlled loading conditions also serves to interpret fatigue testing on structural elements [19,20,21]. In the past, experiments have been carried out regarding the fatigue life of deep beams [22,23,24,25], shear-critical concrete beams [19,26,27,28,29,30,31,32,33,34], and shear-critical slabs [35]. The loading conditions are important for structural tests; research [36] indicates that the fatigue life under moving loads is lower than under cycles of loading applied at a single position. As such, the effect of loading needs to be considered when applying test results to the assessment of existing bridges under traffic loads. Besides the previously mentioned experimental campaigns, experiments on partially prestressed concrete beams showed that the failure mode can change from flexure to shear [20,37,38,39,40,41]. From a practical perspective, fatigue also influences the serviceability behavior of concrete structures, such as two-way reinforced concrete floors [42]. 

For this work, we focus on the relation between the number of cycles of loading and the limit to the concrete compressive strength. This limit is typically expressed as a fraction of the static compressive strength of the concrete, *S_max_*, a value between 0 and 1. In a classic fatigue test of a concrete specimen (most often a cylinder) under compression, the load is applied as a sine wave between fixed lower and upper values. These loads induce stresses in the concrete specimen that fluctuate with *S_min_f_c_* and *S_max_f_c_*. In some experiments, other sequences of loading have been used, including loading with rest periods and using variable amplitude fatigue load testing [43,44,45]. The focus of this work is only on constant amplitude loading. When *S_min_* and *S_max_* are chosen as the input values for an experiment, the outcome of the experiment then is the number of cycles to failure, *N*. 

Usually, the linear relation between the strength degradation (expressed as *S_max_*) and the logarithm of the number of cycles to failure *N* is given, and it is called the Wöhler curve. Such curves can be derived for different values of *S_min_*. For the design of a new structure, we usually know the number of cycles the structure needs to withstand (for example, one million cycles) and carry out the design or the assessment based on the reduced strength associated with this number of cycles. Therefore, in this work, we selected the number of cycles, *N,* as one of the input variables and the reduced strength ratio, *S_max_*, as the output value.

When testing concrete specimens under fatigue compression, a number of parameters can be studied. The most important parameters are *S_min_* and *S_max_*. Mix properties, such as amount of cement, entrained air, water–cement ratio, curing conditions, and age at testing, were found not to be of significant influence on the number of cycles to failure for a given value of *S_max_* [1]. The influence of testing frequency *f* on the fatigue life is a topic of discussion; some authors observed that, for high values of *S_max_*, there is a decrease in fatigue life for a decrease in frequency [1]. For high strength concrete, Hsu [13] came to the opposite conclusion, whereas for ultra-high performance concrete (UHPC), the same observation was made [46]. The influence of the concrete compressive strength is important on the fatigue life. Experimental work [12,47,48] indicated that the fatigue life is reduced for high strength concrete, but no consensus exists on this topic. To remain on the conservative side, older codes prescribe a lower fatigue life for high strength concrete. Fibers were not found to influence the fatigue strength [46,49].

Table 1 gives an overview of some currently and formerly used code equations that are used in this study. NEN 6723:2009 [50] is the Dutch national code for concrete bridges that was replaced by the Eurocodes. This code describes a Wöhler curve for concrete under compression. NEN-EN 1992-1-1+C2:2011 [51] is the general Eurocode for concrete structures. This code checks Equation (7), which is a check for 1 million cycles. The code does not prescribe a Wöhler code. For bridges, NEN-EN 1992-2+C1:2011 [52] checks damage with Equation (11) for any given number of cycles. Finally, the *fib* model code [53] describes a Wöhler curve with two branches, see Equation (18). These equations are used in this study for comparison of our proposed ANN-based expression. 

The strength reduction of concrete subjected to cycles of compression is typically expressed as a function of the number of cycles. In this work, we studied the reduced capacity as a function of a given number of cycles by means of artificial neural networks. Artificial neural networks (ANNs) are a form of machine learning [54] and can be considered the oldest [55] and the most powerful technique [56]. Neural nets have been applied in a wide variety of research fields [57,58], including civil engineering [59,60,61,62,63,64,65,66,67,68,69,70,71]. Their advantage as compared to other modeling techniques such as multi-variate nonlinear regression is that we do not need to estimate the shape of the function a priori [63]. 

ANNs [72] are models that work in the same way as the brain with neurons as processing units. The basic elements of the architecture of a neural net, see Figure 1, are the nodes, the *L* layers in which the nodes are placed, and the transfer function of the neuron, which turns the input of the neuron into the output. The neural nets considered in this work were feedforward—the data presented as input for a layer flowed in the forward direction only. Through optimization algorithms, we found the unknowns of the neural net—the synaptic weight of the connection between every two neurons *W*, and the bias of the neuron *b*, expressed mathematically as *W* and *b* arrays. The optimization algorithm minimizes a performance measure of the network. In our study, the performance measures were the mean error, the maximum relative error, and the percentage of errors larger than 3%. During learning (i.e., following the optimization procedures), the input dataset was subdivided into training, validation, and testing. The training dataset was used for the initial fitting of the model. The validation dataset was used to check the initially derived model and to further optimize the model. Finally, the testing dataset was used to independently check the model without making further changes to it. Early stopping and testing of the proposed neural net avoided overfitting of the data, see Figure 2. Overfitting results in a model that corresponds too closely to the used dataset so that the model has lost its generalizability. 

In this work, we combined an existing database of fatigue experiments [73] with the powerful tool that is a neural network to come to a more accurate description of the fatigue life of concrete specimens under compression. The proposed model is more accurate than the existing code equations, thus it can be used to obtain a better estimate of the fatigue life of concrete elements under compression.

## 2. Materials and Methods 

### 2.1. Data Gathering

We compiled the dataset for the input of the model based on an existing database [73] with 616 experimental results. To have unique input values for the model, we calculated the geometric average of the number of cycles *N* for repeat tests. In addition, we did not include the experimental results on ultra-high performance fiber reinforced concrete (UHPFRC) since we could not ensure a good continuum of input values of the concrete compressive strength, and we removed a few experiments on heat-treated specimens, since it was reported that their fatigue performance was different from regular specimens [46]. We only included the results from experiments with constant amplitude fatigue testing. Variable amplitude fatigue testing is outside the scope of this study; we did not aim to replace the Palmgren–Miner rule for such loading conditions. The dataset included experiments on cylinders, prisms [12,74], and cubes. The result was an input dataset of 203 unique datapoints obtained from references [12,13,46,47,48,49,74,75,76,77,78,79,80,81,82]. The resulting input dataset is available in the public domain [83]. By using the geometric average of the number of cycles to failure *N* of repeat tests, some of the inherent scatter to experimental results was lost. The reader should keep in mind this effect. 

Table 2 shows the input and the output variables in the dataset, as well as the range of values. The number of input parameters was limited, since we wanted to find a model that could use the same input parameters as the code-prescribed models but provide superior accuracy. Figure 3 shows the input and the output variables used in the dataset based on a loading scheme in an experiment. We chose to use *S_max_* as the output value, since this approach was in line with the design procedure of finding the compressive capacity under fatigue. In experiments, the output value was the number of cycles, *N*, but for design, this value was an input based on the required service life of the structure. The number of cycles from *S_min_* to *S_max_* and back (see Figure 3) that can be completed in one second is called the frequency, *f*, which has units [Hz]. The most commonly used frequency is 1 Hz, but the database includes experiments with frequencies ranging from 0.0625 Hz [81] to 65 Hz [49]. We did not select the frequency as an input variable for our model, since the frequency is a property of experiments and not a value used for design or assessment of concrete structures. 

The input values in the database are subdivided into an input value related to the concrete material properties—the concrete compressive strength, *f_c_*. We used the reported average measured values here. As can be seen in Table 2, the dataset encompasses low to very high strength concrete samples. In terms of the loading conditions, the input values were the lower limit of the stress range, *S_min_*, and the number of cycles to failure, *N*. The ranges of parameters in Table 2 show that the dataset included a wide range of values for *S_min_* and that the dataset included low- and high-cycle fatigue tests.

The output of the model was the maximum stress range, *S_max_*. Again, the range of values in Table 2 shows the wide range of stresses covered. 

The dataset was used for training, validation, and testing of the neural network. The percentage of data assigned to each of these tasks is discussed in the next parts. The reader should keep in mind that the developed model is only valid within the ranges of parameters from the input dataset and cannot be extrapolated outside of these parameter ranges. 

### 2.2. Characteristics of Artificial Neural Networks in This Study

In this work, 14 features of the algorithm that finds the optimal ANN were varied, including features of data pre- and post-processing. An overview of these features, which were selected from the literature, is given in Table 3 and Table 4. A description of these features can be found in [59], (note that F14 has been removed), and the way in which the input data is divided into the training, the validation, and the testing subsets (F4) is given in [84]. The work was coded in Matlab [85] using the neural network toolbox for popular learning algorithms [options 1–3 of feature 13 (F13) in Table 4]. The validation of the developed software can be found in [86]. Moreover, several papers involving the successful application of this software have already been published [84,87].

To find the optimal combination of the 14 features, an algorithm that approaches all possible combinations in an optimal way was developed. The full procedure is detailed in [59]. In total, 219 combinations of these features were explored, after which we selected the optimal neural net. 

## 3. Results

### 3.1. Proposed ANN-Based Model

The proposed model was a multi-layer perceptron net (MLPN) with five layers and a distribution of nodes/layer of 3-4-4-4-1, resulting in 12 hidden nodes in total. The network was fully connected, and the hidden as well as the output transfer functions were all Hyperbolic Tangent and Identity, respectively. The network was trained using the Levenberg–Marquardt (LM) algorithm (1500 epochs). After design, the average network computing time for a single example (including data pre-/post-processing) was 7.09 × 10^−5^ s. Figure 4 depicts a simplified scheme of some of the network key features. The max error was 5.1%, performance of all data was 1.2%, and the percentage of errors larger than 3% was 10.3% based on the original input and output values (before normalization and dimensional analysis). The properties of the microprocessor used in this work were OS: Win10Home 64 bits; RAM: 48 GB; Local Disk Memory: 1 TB; CPU: Intel® Core™ i7 8700 K @ 3.70–4.70 GHz.

The input data were a vector of three components, *Y_1_*. The input vector *Y_1_* contained *f_c,cyl_*, *S_min_*, and *N*, as shown in Table 2. After input normalization, the new input dataset ${\left\{Y_{1}\right\}}_{n}^{after}$ was defined as: (23)$$\begin{array}{l} {\left\{Y_{1,sim}\right\}}_{n}^{after}\;=\;\left({\left\{Y_{1,sim}\right\}}_{d.r}^{after}\;-\;INP\left(:,1\right)\right)\;./\;INP\left(:,2\right)\\ INP=\left[\begin{array}{cc} 70.7417339901478 & 44.7805334345334\\ 0.206873201970443 & 0.192556137441602\\ 1122471.30034975 & 5728851.48603107 \end{array}\right] \end{array}$$
where one recalls that operator ‘./’ divides row *i* in the numerator by INP(*i*, 2). 

Once we determined the preprocessed input dataset {*Y*_1_}*_n_^after^* (3 × 1 matrix), the next step was to present it to the proposed ANN to obtain the predicted output dataset *Y_5_* (single value, *S_max_* as shown in Table 2). 

Next, the mathematical representation of the proposed ANN is given so that any user can implement it to determine *Y_5_*, thus eliminating all rumors that ANNs are “black boxes”.
(24)$$\begin{array}{l} Y_{2}\;=\varphi_{2}\left(W_{1-2}^{T}{\left\{Y_{1,sim}\right\}}_{n}^{after}+b_{2}\right)\\ Y_{3}\;=\varphi_{3}\left(W_{1-3}^{T}{\left\{Y_{1,sim}\right\}}_{n}^{after}+W_{2-3}^{T}Y_{2}+b_{3}\right)\\ Y_{4}\;=\varphi_{4}\left(W_{1-4}^{T}{\left\{Y_{1,sim}\right\}}_{n}^{after}+W_{2-4}^{T}Y_{2}+W_{3-4}^{T}Y_{3}+b_{4}\right)\\ {\left\{Y_{5,sim}\right\}}_{n}^{after}\;=\varphi_{5}\left(W_{1-5}^{T}{\left\{Y_{1,sim}\right\}}_{n}^{after}+W_{2-5}^{T}Y_{2}+W_{3-5}^{T}Y_{3}+W_{4-5}^{T}Y_{4}+b_{5}\right) \end{array}$$
where
(25)$$\begin{array}{l} \varphi_{2}=\varphi_{3}=\varphi_{4}=\varphi \;(s)=\frac{e^{s}-e^{-s}}{e^{s}+e^{-s}}\\ \varphi_{5}=\varphi_{5}(s)=s \end{array}$$

Arrays *W_j-s_* and *b_s_* are stored online in [88].

### 3.2. Performance Indicators of Results

The obtained ANN solution for every data point can be found in [83], making it possible to compute the exact (with all decimal figures) approximation errors. All results were calculated based on effective target and output values, i.e., computed in their original format. This proposed model resulted in a maximum relative error of 5.1%, 10.3% of predictions with an error larger than 3%, and a mean relative error of 1.2% for the 203 datapoints. For the training subset, the mean relative error was 1.3%, for the validation subset, 1.1%, and for the testing subset, 1.2%. The relative error *e* is defined as follows:(26)$$ℯ=100\left|\frac{S_{max,test}-S_{max,ANN}}{S_{max,t}}\right|$$
with *S_max,test_* the experimental value of *S_max_* and *S_max,ANN_* the predicted value based on the neural network. Figure 5 shows the tested versus the predicted values for the ANN-based model for each datapoint as well as the *R*-value, which was 0.99238 for this case. 

### 3.3. Comparison between ANN-Based and Existing Methods

To highlight the advantage of using the proposed ANN-based model, we compare the proposed model to the existing code models from Table 1 in this section. The result for each datapoint as calculated with the proposed model is available for download [83]. To calculate the predicted values for *S_max_* with the code formulas, we used average values for the concrete compressive strength instead of design or characteristic values for the calculation of *f’_b,rep,v_*, *f_cd,fat_* and *f_ck,fat_* but not for the correction terms of *f_ck_/*250 MPa and *f_ck_/*400 MPa in the Eurocode and the Model Code expressions, respectively. Here, *f_ck_* was determined as *f_c,avg_*—8 MPa [51]. Figure 6 shows the comparison between the tested and the predicted values. Note that the values of NEN-EN 1992-1-1+C2:2011 [51] were not included, since these are only for one million cycles and were thus not applicable to most of the datapoints in our input dataset. We can also see that a few datapoints gave a negative value for the predicted *S_max_*, which was, of course, physically not possible. In total, seven datapoints calculated with NEN-EN 1992-2+C1:2011 [52] gave a negative value, and one point with NEN 6723:2009 [50] resulted in a negative value. The expression from NEN-EN 1992-2+C1:2011 [52] was not developed for high strength concrete. As such, for datapoints with a high compressive strength and/or a high value of *S_min_*, we could not calculate a value of *S_max_* that was larger than *S_min_*_,_ and the expressions resulted in a negative solution, which was physically not possible. This effect was more pronounced for high strength concrete because of Equation (5), where the reduction term *f_ck_/*250 increased for increasing concrete compressive strengths. From a computational point of view, we note that taking the log of both sides of Equation (12) resulted in more stable results for the outcome of *S_max_* when using the MathCad 15.0 [89] solver to find *S_max_* for given input values of *f_c,avg_*, *S_min_*, and *N*. The one datapoint that resulted in negative values with NEN-EN 1992-2+C1:2011 [52] and NEN 6723:2009 [50] had *f_c,avg_* = 41 MPa (normal strength concrete) and *S_min_* = 0.836. Since the value of *S_min_* was very high, the expressions could not find a solution for *S_max_* larger than *S_min_*^,^ giving a negative value instead. We should remark as well that the expressions of *fib* model code 2010 [26] always resulted in a value for *S_max_* but that, for cases where *S_min_* was large, the calculated value for *S_max_* could be smaller than *S_min_*, which was also physically not possible. 

Table 5 gives the statistical properties of the ratio of tested to predicted values with the code equations from Table 1 and with our proposed model. There are two rows with results for the code equations—the first row per code gives the statistical properties for all datapoints, and the second row gives the properties only for the datapoints where the calculation was physically possible, i.e., we removed the datapoints where we found a negative value of *S_max_* or a value of *S_max_* smaller than *S_min_*. From this analysis, we can see that the expressions of the *fib* model code [53] led to the best results of the expressions in Table 1 with an average tested to predicted value of 1.37 and the coefficient of variation equal to 20.5%. We can also observe that our proposed model led to a better prediction with an average tested to predicted value of 1.00 and a coefficient of variation of 1.7%. 

## 4. Discussion

From the results presented in Section 3.3, we can see that our proposed model was a significant improvement with respect to the existing code equations. The ANN-based proposed model used the data from the literature in an optimal way and led to good results because sufficient experimental results were available. The model we developed can thus be used for the design and the assessment of concrete structures and provides a more accurate assessment and design than the existing methods. We need to remark here, however, that the ANN-based model predicts average values. While further statistical studies would be necessary to derive a design matrix-based formulation from this approach, we suggest the use of *f_cd_* instead of *f_c,avg_* as an input value and the use of *S_min_* resulting from the serviceability limit state load combination. The reason for the latter recommendation is that the experimental results indicate that the fatigue life increased as *S_min_* increased, thus it would be a more conservative approach to use the load combination that results in the lowest value for *S_min_*. 

In Section 2.1, we explained the data gathering process. The reader should remember that, for literature sources where repeat experiments were reported, we used the geometric average of the experimental results. This approach is in line with literature references where not all results from all experiments were reported but instead only the geometric average of the experimental results. However, this approach removed some of the inherent scatter on the experimental results from the input database. The reader should keep this restriction in mind.

As with every ANN-based model, this model is only valid for the ranges of parameters of the input dataset, as given in Table 2. This limitation is the main disadvantage of the proposed model. However, we can see in Table 2 that the ranges of parameters in our input dataset were quite large. The input dataset included experiments with relatively large values of *S_min_* up to 0.836. As seen in Figure 6, the existing code formulas could not predict the value for *S_max_* when *S_min_* was larger. While such cases are uncommon in practice, it is an advantage of our proposed ANN-based model that this model can address cases with a large value for *S_min_* as input. 

The input dataset included experiments with high strength concrete with steel fibers in the mix with a maximum concrete compressive strength of 170 MPa. In the range of concrete compressive strengths from 24 MPa to 170 MPa, we could ensure a fairly continuous increase in values of the concrete compressive strength. We included specimens with steel fibers in the concrete mix, since [46] showed that the fibers do not influence the fatigue strength of concrete under compression. We did not include ultra-high performance fiber reinforced concrete (UHPFRC) specimens, as reported in [90], nor the heat-treated specimens from [46]. The UHPFRC specimens were omitted since we could not achieve good continuous increases in the concrete compressive strength for the largest ranges of the concrete compressive strength. The heat-treated specimens were omitted since their observed behavior was different from regular specimens. However, if more experiments on heat-treated specimens would become available, the study presented in this paper could be repeated with the additional input parameter “heat-treated (yes/no)”. Note that the algorithm that searches for the optimal neural network can process both quantitative and qualitative data. Similarly, if more experimental data in the UHPFRC range of concrete compressive strength would become available, this information could be added to the database, and the study could be repeated. 

Given the earlier discussions on the fatigue life of high strength concrete [73], we found it valuable to include high strength concrete specimens in our study. As such, our study was also an improvement with respect to the existing code formulas. In particular, we can see in Figure 6 that expressions from NEN-EN 1992-2+C1:2011 [52] were not suitable for predicting *S_max_* for high strength concrete, as physically impossible values for *S_max_* were found. 

The input database included low- and high-cycle fatigue tests with the number of cycles to failure ranging from three to almost 64 million. Low-cycle fatigue [16,91] can be interesting for practical cases where we want to study the fatigue life of a bridge member subjected to a limited number of very heavily loaded trucks. High-cycle fatigue [22,34,46], on the other hand, is interesting for two reasons; for regular design and assessment, we assumed a number of cycles of 250 million to 500 million cycles [3], and with high-cycle fatigue, we could study the so-called fatigue limit [92], i.e., the number of cycles for which *S_max_* did not decrease further in the Wöhler curve. The fatigue limit in concrete is, however, subject to discussion. While the dataset had a maximum value of *N* close to 64 million cycles, there are no experimental results available that cover the range of 250 million cycles and higher. The reason why such experimental results are not available is that the amount of time needed becomes very large. For example, say we want to test 500 million cycles with a loading frequency of 1 Hz, as is commonly used in fatigue testing. Such an experiment would take 500 million seconds, or close to 16 years, to complete. Then, given the large scatter inherent in fatigue testing, we would need to repeat the experiment a number of times, which would only add to the time required for obtaining these test data. It would, however, be interesting to have such data available—not only for studying the fatigue limit and the number of cycles used for design and assessment but also to study the code formulas of the *fib* Model Code 2010 [53], which defines a change in Wöhler curve for 100 million cycles. Given the considerations in the previous paragraphs, the input database and the resulting proposed model form an improvement with respect to the existing code formulas.

To further study the influence of the parameters on the code expressions and our proposed model, we studied the relation between each parameter and the ratio of tested to predicted values for *S_max_*. The first parameter analyzed was *S_min,_* see Figure 7. We can see in this figure that the ANN-based model performed consistently over the full range of values of *S_min_*. We remarked earlier that we could not find a solution for NEN 6723:2009 [50] for the datapoint with the largest value of *S_min_* and that various datapoints with a large value of *S_min_* did not lead to a physically possible solution with the expressions from NEN-EN 1992-2+C1:2011 [52]. The values of the tested to predicted *S_max_* seemed to slightly decrease as *S_min_* increased for the predictions with the *fib* Model Code 2010 [53]. 

The second parameter to further analyze was the concrete compressive strength. Figure 8 shows the relation between the average concrete compressive strength and the ratio of tested to predicted value of *S_max_*. We can observe from this plot that our proposed model performed equally well over the full range of concrete compressive strengths. We can see that the expressions from NEN-EN 1992-2+C1:2011 [52] were overly conservative. However, we need to keep in mind that C90/105 is the highest strength concrete class in NEN-EN 1992-1-1:2005 [93], thus some high strength concrete specimens in our dataset were outside the scope of the Eurocodes. In particular, the term *f_ck_*/250 MPa in Equation (5) was overly conservative for high strength concrete. We can see in Figure 8 that the *fib* Model Code term of *f_ck_/*400 MPa from Equation (13) led to better results from high strength concrete. Figure 8 also shows that the predictions for *S_max_* were still more conservative for high strength concrete than for normal strength concrete. In that regard, the expressions from NEN 6723:2009 [50] seemed to have a more uniform performance over the full range of concrete compressive strengths. 

The next studied parameter was the number of cycles, *N*, shown as log*N* in Figure 9, where the tested to predicted ratios of *S_max_* are shown as a function of log*N*. We can see that the code equations were less conservative for low-cycle fatigue than for high-cycle fatigue. This observation was stronger for NEN 6723:2009 [50] than for the other codes. From experiments [48], we know that the Wöhler curve starts to be linear after 100 cycles. As such, it was expected that the datapoints for log*N* ≤ 2 would be more difficult to predict. Again, our proposed model performed well over the full range of cycles in the input dataset.

The last studied parameter was *S_max_* itself. Figure 10 shows the ratio of tested to predicted ratios of *S_max_* as a function of *S_max_*. We can see from this plot that the code predictions tended to become more conservative as *S_max_* increased, whereas our proposed model performed well and consistently over the full range of values of *S_max_* in the input dataset.

Finally, we explored if there was a difference between the Wöhler curve resulting from the experimental results and from the ANN-based predictions. Figure 11 shows these results and the Wöhler curves. The reader can observe that the difference between the two Wöhler curves was minimal.

As compared to previously developed ANN-based expressions for similar problems in structural concrete, e.g., problems where the amount of experimental data is large but the theoretical understanding is limited, we found larger errors for this problem. Other structural concrete problems that we studied with a similar approach were the shear capacity of one-way slabs without shear reinforcement [59] and the shear capacity of steel fiber reinforced concrete beams without stirrups [84]. These observations are in line with the scatter observed in experiments. 

The model we propose herein is a relatively simple and easy to use model. We used only three input variables to stay in line with the currently used code formulations. The computational time per datapoint is very fast, 7.09× 10^−5^ s per datapoint. Since we provided all expressions for the readers in this work and the W and the b arrays in the public domain, direct implementation of our proposed model is easy. The reader can set up a spreadsheet with the equations from Section 3.1 and from then on can use our proposed model quickly and easily. This observation again underlines the improvement of our proposed model with respect to existing models. 

Our proposed model did not explain the mechanics that drive fatigue failure of concrete under compression. Research on this topic is still necessary, and mechanics-based models are necessary. However, the currently available code equations do not perform very well when compared to experimental results. Therefore, better expressions, such as our proposed model, can be used until mechanics-based expressions (with limited scatter) are available. Until then, our proposed model can be a useful tool for the design and the analysis of concrete structures in a more efficient and cost-effective way.

## 5. Summary and Conclusions

We proposed, in this paper, the use of an ANN-based model for the determination of the concrete compressive strength for specimens subjected to cycles of loading based on experimental results from the literature. We derived the expression as follows:We derived an input dataset with 203 datapoints obtained from experiments reported in the literature. Each datapoint was unique. Where necessary, the geometric average of number of cycles to failure of repeat tests was determined;We selected three input parameters for the input dataset (concrete compressive strength, lower bound of the stress ratio, and number of cycles to failure) and one output parameter (upper bound of the stress ratio) in line with the parameters used in the currently used code expressions;We used different methods for 14 features of the ANN models to find the most suitable features. We looked at 219 combinations of features and selected the neural net with the best performance.

The main advantages of our proposed approach and the main findings of this study are as follows:Of the studied methods in the current codes, we found that the expression from the *fib* model code performed best when compared to the experimental results gathered in the dataset. The average value of tested to predicted upper bound of the stress ratio was 1.37 with a coefficient of variation of 20.5%;We can see that our proposed model outperformed the code equations for the prediction of the upper bound of the stress ratio, since the average value of tested to predicted upper bound of the stress ratio was 1.00 with a coefficient of variation of 1.7%;The tested to predicted values obtained with our proposed model did not show any dependence on any of the input or the output parameters, i.e., our model performed consistently well over the full range of the parameters. In contrast, plotting the tested to predicted ratios obtained with the code equations showed that these depended on the input and the output parameters. In particular, the predicted values for the upper bound of the stress ratio with the code equations became overly conservative as the concrete compressive strength increased;The computational time of our proposed neural net is small (0.07 milliseconds per datapoint).

The limitations of our proposed model and the necessities for future work are:Experiments on high-cycle fatigue are necessary. Given the required time for such experiments, however, it is unlikely that such experiments can be carried out. Perhaps, numerical analyses can be used to generate datapoints for *N* > 64 million cycles;The proposed method was derived from test results and aims at average values. Further studies are necessary to define the safety factors for design and assessment based on our proposed method;The proposed ANN-based model is only valid for the parameter ranges in the input dataset. However, these ranges cover most practical cases, except, as mentioned earlier, *N* > 64 million cycles. In practice, we need 250 or 500 million cycles for design and assessment;The mechanics of the problem and the reasons for the large scatter in fatigue tests were not addressed in this study. The parameter studies presented here, however, give insight in the governing parameters and can be used in the future for comparison to expressions that are derived based on mechanics.Regardless the high quality of the predictions yielded by the proposed model for the used data, the reader should not blindly accept that model as accurate for any other instances falling inside the input domain of the design dataset. Any analytical approximation model must undergo extensive validation before it can be taken as reliable (the more inputs, the larger the validation process). Models proposed until that stage are part of a learning process towards excellence.

## Figures and Tables

**Figure 1 materials-12-03787-f001:**
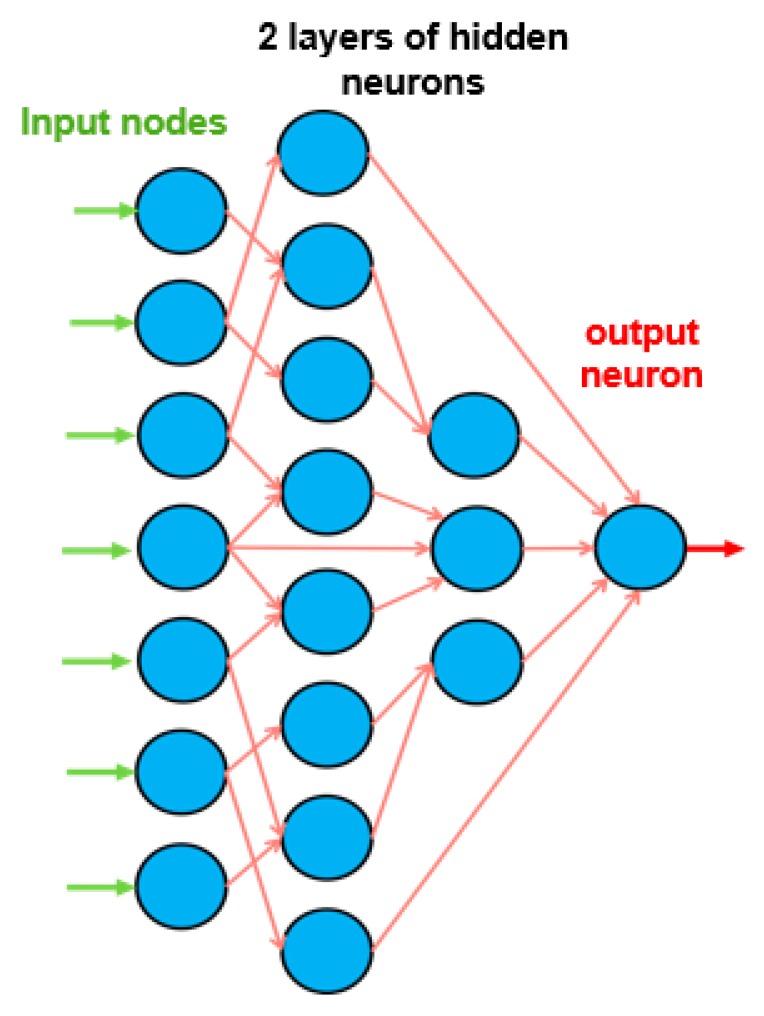
Example of feedforward neural network.

**Figure 2 materials-12-03787-f002:**
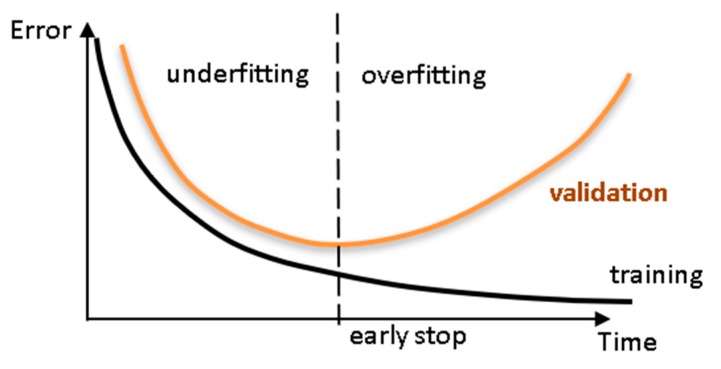
Cross-validation—assessing network’s generalization ability.

**Figure 3 materials-12-03787-f003:**
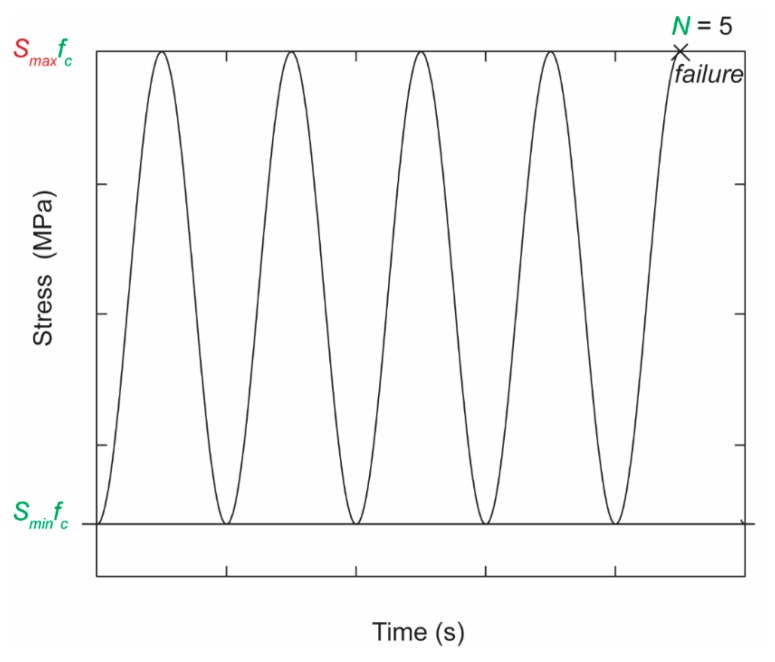
Input (green) and output variables (red), shown on an example of a loading scheme in an experiment.

**Figure 4 materials-12-03787-f004:**
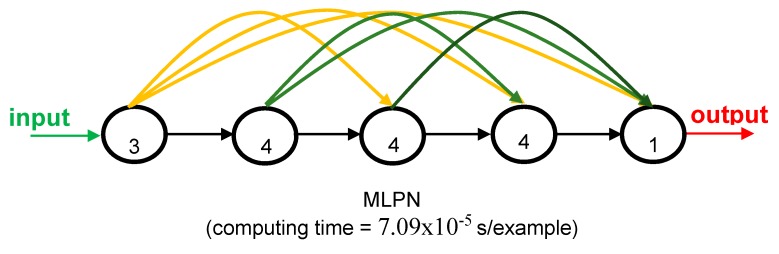
Proposed 3-4-4-4-1 fully connected MLPN—simplified scheme.

**Figure 5 materials-12-03787-f005:**
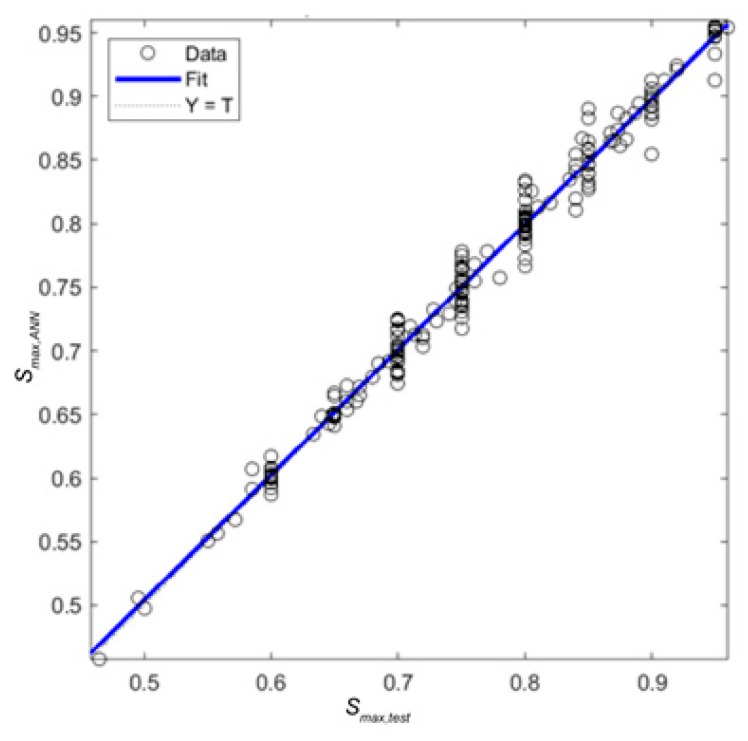
Regression plot for the proposed ANN for the output variable, S_max_. The expression for the blue line is: *S_max,ANN_* = 0.98 *S_max,test_* + 0.012 and *R* = 0.99238.

**Figure 6 materials-12-03787-f006:**
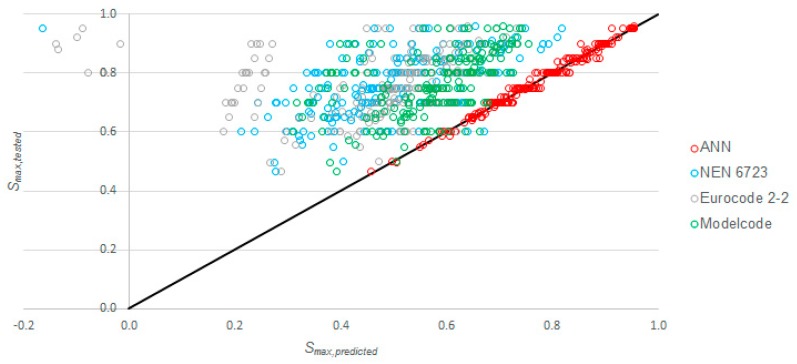
Comparison between tested and predicted values with code formulas and ANN-based model.

**Figure 7 materials-12-03787-f007:**
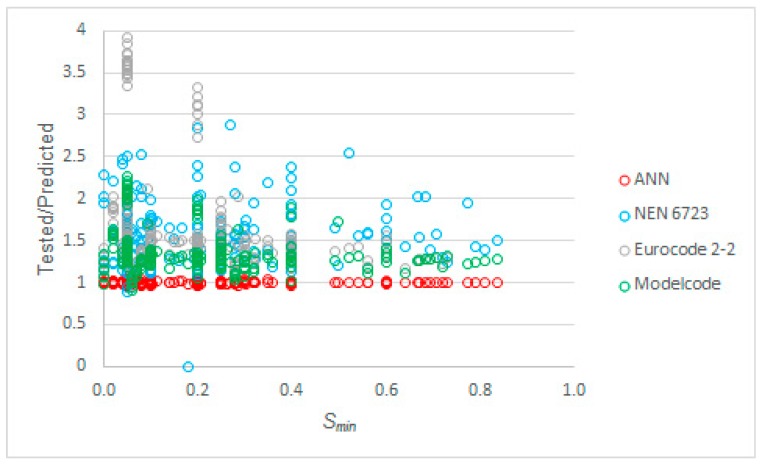
Tested to predicted values for all considered methods as a function of *S_min._*

**Figure 8 materials-12-03787-f008:**
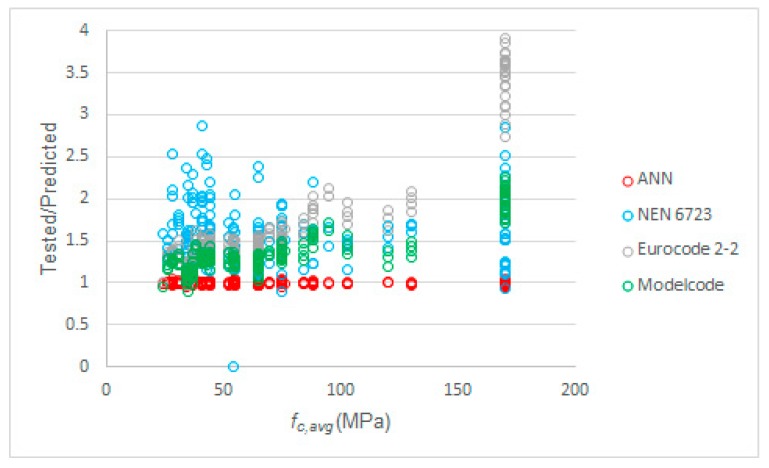
Tested to predicted values for all considered methods as a function of the average concrete compressive strength.

**Figure 9 materials-12-03787-f009:**
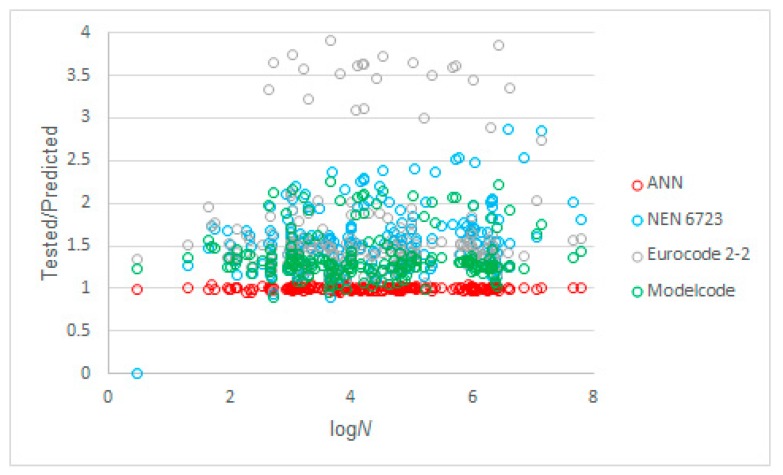
Tested to predicted values for all considered methods as a function of the number of cycles to failure, *N*.

**Figure 10 materials-12-03787-f010:**
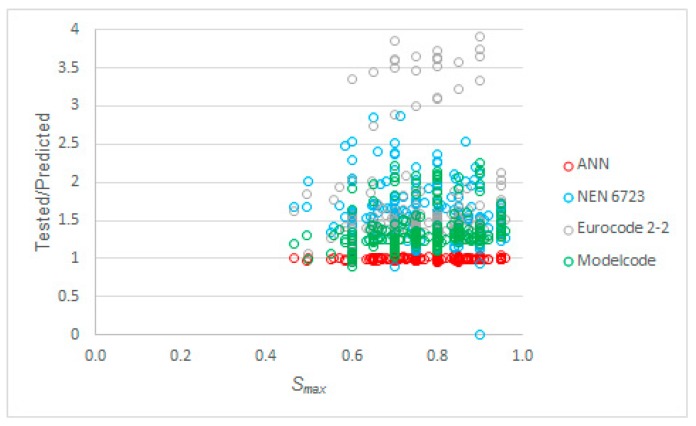
Tested to predicted values for all considered methods as a function of *S_max_*.

**Figure 11 materials-12-03787-f011:**
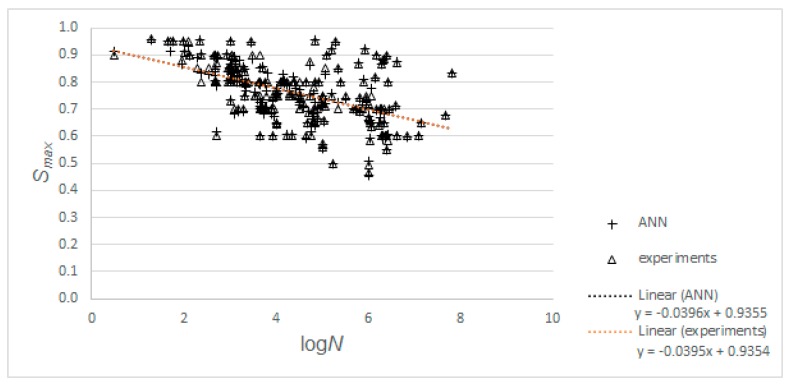
Comparison between Wöhler curve resulting from experimental data and from ANN-based predictions.

**Table 1 materials-12-03787-t001:** Overview of code expressions for fatigue.

Code	Ref	Equations	Nr
NEN 6723:2009	[50]	$f_{b,v}^{\prime }=\frac{f_{b,rep,v}^{\prime }}{\gamma_{m}}$ with *γ_m_* = 1.2	(1)
		$f_{b,rep,v}^{\prime }=0.5\left(f_{b,rep,k}^{\prime }-0.85\times 30\right)+0.85\times 30\;\mathrm{in}\;[\mathrm{MPa}]$	(2)
		$Log\left(N\right)=\frac{10}{\sqrt{1-R}}\left(1-\frac{\sigma \prime_{b,d,max}}{f\prime_{b,v}}\right)\;\mathrm{for}\;\frac{\sigma \prime_{b,d,max}}{f\prime_{b,v}}>0.25$	(3)
		$R=\frac{\sigma_{b,d,min}^{\prime }}{\sigma_{b,d,max}^{\prime }}=\frac{S_{min}}{S_{max}}$	(4)
NEN-EN 1992-1-1+C2:2011	[51]	$f_{cd,fat}=k_{1}\beta_{cc}(t_{0})f_{cd}\left(1-\frac{f_{ck}}{250}\right)$ with *f_ck_* in MPa and *k_1_* = 0.85	(5)
		$\beta_{cc}(t_{0})=\exp\left\{s\left[1-{\left(\frac{28}{t_{0}}\right)}^{0.5}\right]\right\}$	(6)
		$E_{cd,max,equ}+0.43\sqrt{1-R_{equ}}\leq 1$	(7)
		$R_{equ}=\frac{E_{cd,min,equ}}{E_{cd,max,equ}}$	(8)
		$E_{cd,max,equ}=\frac{\sigma_{cd,max,equ}}{f_{cd,fat}}$	(9)
		$E_{cd,min,equ}=\frac{\sigma_{cd,min,equ}}{f_{cd,fat}}$	(10)
NEN-EN 1992-2+C1:2011	[52]	$\sum_{i=1}^{m}\frac{n_{i}}{N_{i}}\leq 1$	(11)
		$N_{i}=10^{\left(14\frac{1-E_{cd,max,i}}{\sqrt{1-R_{i}}}\right)}$	(12)
*fib* model code 2010	[53]	$f_{ck,fat}=\beta_{cc}(t)\beta_{c,sus}(t,t_{0})f_{ck}\left(1-\frac{f_{ck}}{400}\right)$ with *f_ck_* in MPa, *β_c,sus_*(*t, t_0_*) = 0.85 and *s* = 0.25 for cement class 42.5 N	(13)
		$\beta_{cc}(t)=\exp\left\{s\left[1-{\left(\frac{28}{t}\right)}^{0.5}\right]\right\}$	(14)
		$t_{T}=\sum_{i=1}^{n}\Delta t_{i}\exp\left(13.65-\frac{4000}{273+T(\Delta t_{i})}\right)$	(15)
		$\log N_{1}=\frac{8}{Y-1}\left(S_{c,max}-1\right)$	(16)
		$\log N_{2}=8+\frac{8\ln(10)}{Y-1}\left(Y-S_{c,min}\right)\log\left(\frac{S_{c,max}-S_{c,min}}{Y-S_{c,min}}\right)$	(17)
		$\log N=\left\{ \begin{array}{l} \log N_{1}\;\mathrm{if}\;\log N_{1}\leq 8\\ \log N_{2}\;\mathrm{if}\;\log N_{1}>8 \end{array}\right.$	(18)
		$Y=\frac{0.45+1.8S_{c,min}}{1+1.8S_{c,min}-0.3S_{c,min}^{2}}$	(19)
		$S_{c,max}=\frac{\left|\sigma_{c,max}\right|}{f_{ck,fat}}$	(20)
		$S_{c,min}=\frac{\left|\sigma_{c,min}\right|}{f_{ck,fat}}$	(21)
		$\Delta S_{c}=|S_{c,max}|-|S_{c,min}|$	(22)

**Table 2 materials-12-03787-t002:** Overview of input and output variables in the dataset, including ranges of values.

Input Parameters			Input Number	Min	Max
**Concrete properties**	***f_c,cyl_* (MPa)**	average concrete compressive strength	1	24	170
**Loading**	***S_min_* (-)**	lower limit of stress range	2	0	0.836
	***N* (-)**	number of cycles to failure	3	3	63,841,046
**Output**	***S_max_* (-)**	upper limit of stress range	1	0.465	0.960

**Table 3 materials-12-03787-t003:** Implemented artificial neural network (ANN) features (F) 1–7. The highlighted cells show the features that were used to derive the final neural net.

F1	F2	F3	F4	F5	F6	F7
Qualitative Var Represent	Dimensional Analysis	Input Dimensionality Reduction	% Train-Valid-Test	Input Normalization	Output Transfer	Output Normalization
Boolean Vectors	Yes	Linear Correlation	80-10-10	Linear Max Abs	Logistic	Lin [a, b] = 0.7[φ_min_, φ_max_]
Eq Spaced in ]0,1]	No	Auto-Encoder	70-15-15	Linear [0, 1]	-	Lin [a, b] = 0.6[φ_min_, φ_max_]
-	-	-	60-20-20	Linear [−1, 1]	Hyperbolic Tang	Lin [a, b] = 0.5[φ_min_, φ_max_]
-	-	Ortho Rand Proj	50-25-25	Nonlinear	-	Linear Mean Std
-	-	Sparse Rand Proj	-	Lin Mean Std	Bilinear	No
-	-	No	-	No	Compet	-
					Identity	

**Table 4 materials-12-03787-t004:** Implemented ANN features (F) 8–14. The highlighted cells show the features that were used to derive the final neural net.

F8	F9	F10	F11	F12	F13	F14
Net Architectue	Hidden Layers	Connectivity	Hidden Transfer	Parameter Initialization	Learning Algorithm	Training Mode
MLPN	1 HL	Adjacent Layers	Logistic	Midpoint (W) + Rands (b)	BP	Batch
RBFN	2 HL	Adj Layers + In-Out	Identity-Logistic	Rands	BPA	Mini-Batch
-	3 HL	Fully-Connected	Hyperbolic Tang	Randnc (W) + Rands (b)	LM	Online
-	-	-	Bipolar	Randnr (W) + Rands (b)	ELM	-
-	-	-	Bilinear	Randsmall	mb ELM	-
-	-	-	Positive Sat Linear	Rand [−*Δ*, *Δ*]	I-ELM	-
-	-	-	Sinusoid	SVD	CI-ELM	-
			Thin-Plate Spline	MB SVD	-	
			Gaussian	-	-	
			Multiquadratic	-	-	
			Radbas	-	-	
			Thin-Plate Spline	MB SVD	-	

Abbreviations: MLPN = multi-layer perceptron net, RBFN = radial basis function net, SVD = singular value decomposition, MB SVD = mini-batch SVD, BP = back propagation, BPA = back propagation with adaptive learning rate, LM = Levenberg–Marquardt, ELM = extreme learning machine, mb ELM = mini-batch ELM, I ELM = incremental ELM, CI ELM = convex incremental ELM, NNC = neural network composite.

**Table 5 materials-12-03787-t005:** Statistical properties of *V_utot_*/*V_pred_* for all datapoints with AVG = average of *V_utot_*/*V_pred_*, STD = standard deviation on *V_utot_*/*V_pred_*, and COV = coefficient of variation of *V_utot_*/*V_pred_*.

Model		AVG	STD	COV	Min	Max
Proposed model		1.00	0.02	1.69%	0.955	1.053
NEN 6723:2009	[50]	1.55	0.63	40.53%	−5.828	2.869
		1.59	0.35	22.27%	0.893	2.869
NEN-EN 1992-2+C1:2011	[52]	1.07	4.59	430.61%	−56.25	3.913
		1.70	0.69	40.90%	0.971	3.913
*fib* model code 2010	[53]	1.37	0.28	20.46%	0.906	2.261
		1.37	0.28	20.68%	0.906	2.261

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
