# Peer review of "ANN-Based Fatigue Strength of Concrete under Compression"

_materials, 2019, doi:10.3390/ma12223787_

Round 1

Reviewer 1 Report

Interesting study. Nonetheless, the main conclusion of the paper, which is that the proposed neural networks model improves exiting formulas (e.g. model code) does not seem to be properly supported. For a given set of data, it is very normal that a model that has been calibrated based on that specific dataset improves the predictions of any other model. This is very normal, since the first model has been calibrated for those specific data while the second model was calibrated based on another set of data. Very likely, the same outcome would have been obtained even if a much simpler formula had been used. In summary:

Whether or not the calibrated model improves existing formulas can only be concluded if the validation database covers all spectrum of concretes, testing procedures, and testing conditions. Nonetheless, the paper does not include any information in this regard. Whether or not the ANN improves a simple equation has not been studied. Is the use of ANN worth it? Any attempt to use a simple equation for the calibration? What was the outcome? There are only three inputs, so the calibration of a simple equation does not seem like an unreasonable alternative.

General remarks:

The paper requires some English grammar editing. The use of first person is not recommended in scientific publications. Instead of “We determined X and Y…”, you may state “X and Y were determined…” Check pdf file for figures calls and numbering. Try to make the paper understandable for someone that does not know much about ANN. Literature review should be improved by including the outcome of more fatigue experimental studies.

Specific remarks:

In abstract: Vtest/VANN is not defined. Line 57 “Mix properties… were found not to be of significant influence”. I am guessing this statement considers that the input variable is stress divided compressive strength, not the stress itself. Please clarify. Line 93. Please explain what “training, validation, and testing” means. The same for “overfitting”. These concepts may be misleading to someone without ANN expertise. Section 2.2. The description of features, feature method, sub-analysis, combo, best combo, etc. is very confusing. My suggestion: 1) merge tables 3, 4, and 5 by including, for each of the features, which are the options you analyzed; 2) highlight, in the same table, which is the option you finally selected for each feature; 3) no need to refer to feature method number, sub-analyses, or combo. Section 3.1. ANN formulae: What does the subindex “sim” stands for? Input normalization explanation is not fully clear. I understand each normalized input is the input minus the mean, divided by the std. If that is the case, why not just stating Yi,norm. = (Yi- ai)/bi ? What is Psim? Is it 1? In that case, why not just stating that, for example, Y1 is a vector of 3 components, or Y5 is just a number? Section 3.2. What is "mean relative error"? Figure 5. What is target and output? Do you mean predicted / measured? Why including the blue line equation as a title for the y-axis? Suggest get rid of Figures 6 and 7 and explain the outcome (it’s only a few numbers) in the text I suggest you include a summary plot where you can show the influence of the three inputs on Smax. You may summarize your model sensitivity in a Wöhler plot.

Reviewer 2 Report

See the attached document, which is the review for the Authors.

Reviewer 3 Report

Good paper - the subject is of significant scientific and potentially practical importance - the strength reduction of concrete subjected to cycles of compression, typically expressed as a function of the number of cycles is considered.

The key elements are present: abstract, introduction, methods, results, discussion/conclusions. The use of an ANN-based model for the determination of the concrete compressive strength for specimens subjected to cycles of loading, based on experimental results from the literature is adequately selected and matches well with the problem, giving more scientific valour to this contribution. The proposed model is more accurate than the existing code 97 equations, so that it can be used to obtain a better estimate of the fatigue life of concrete elements under compression. The limitations of proposed model and necessities for future work are pointed out.

The text seems to be quite clear and well written.

The quality and resolution of the figures and tables (equations) is sufficient.

Author Response

Reviewer 3

Good paper - the subject is of significant scientific and potentially practical importance - the strength reduction of concrete subjected to cycles of compression, typically expressed as a function of the number of cycles is considered.

Thank you for your review. We have prepared a revised version of the manuscript based on the comments of the reviewers.

The key elements are present: abstract, introduction, methods, results, discussion/conclusions. The use of an ANN-based model for the determination of the concrete compressive strength for specimens subjected to cycles of loading, based on experimental results from the literature is adequately selected and matches well with the problem, giving more scientific valour to this contribution. The proposed model is more accurate than the existing code 97 equations, so that it can be used to obtain a better estimate of the fatigue life of concrete elements under compression. The limitations of proposed model and necessities for future work are pointed out.

 The text seems to be quite clear and well written.

The quality and resolution of the figures and tables (equations) is sufficient.

We appreciate your positive evaluation of our work, and hope you agree with the changes and clarifications added to the revised version of the manuscript.

Round 2

Reviewer 1 Report

Reviewer's comments adequately addressed.

Reviewer 2 Report

The Authors have performed a good job. They considered how I had commented their article and suitably addressed all my comments. Not only did they carefully and correctly addressed the issues raised in my review, but also they blended my suggestions or criticisms and further personal developments or contributions, which have further enriched the article.

Now, the article adds to the subject and the presentation saves the readers’ effort to understand the message that the article aims at conveying.

Thus, I recommend that the revised version of the article that has been resubmitted is accepted and published in the present form.